# Insect-Microorganism Interaction Has Implicates on Insect Olfactory Systems

**DOI:** 10.3390/insects13121094

**Published:** 2022-11-28

**Authors:** Shupei Ai, Yuhua Zhang, Yaoyao Chen, Tong Zhang, Guohua Zhong, Xin Yi

**Affiliations:** 1Key Laboratory of Crop Integrated Pest Management in South China, Ministry of Agriculture, South China Agricultural University, Guangzhou 510642, China; 2Key Laboratory of Natural Pesticide and Chemical Biology, Ministry of Education, South China Agricultural University, Guangzhou 510642, China; 3Guangdong Province Key Laboratory of Microbial Signals and Disease Control, College of Plant Protection, South China Agricultural University, Guangzhou 510642, China

**Keywords:** microorganism-insect interactions, insect olfactory, insect behavior, the olfactory system

## Abstract

**Simple Summary:**

Interactions between insects and microorganisms facilitate the spread of the microorganisms, which in turn affect insect behaviors. In the current paper, we reviewed how fungi, bacteria, and viruses act on insect olfaction, ranging from volatiles of microorganism, microorganism-induced changes in host volatiles, and symbiotic microorganisms that directly affect insect olfaction. Finally, we highlighted the importance of olfaction in insect-microorganism interactions and provided insights into future research directions.

**Abstract:**

Olfaction plays an essential role in various insect behaviors, including habitat selection, access to food, avoidance of predators, inter-species communication, aggregation, and reproduction. The olfactory process involves integrating multiple signals from external conditions and internal physiological states, including living environments, age, physiological conditions, and circadian rhythms. As microorganisms and insects form tight interactions, the behaviors of insects are constantly challenged by versatile microorganisms via olfactory cues. To better understand the microbial influences on insect behaviors via olfactory cues, this paper summarizes three different ways in which microorganisms modulate insect behaviors. Here, we deciphered three interesting aspects of microorganisms-contributed olfaction: (1) How do volatiles emitted by microorganisms affect the behaviors of insects? (2) How do microorganisms reshape the behaviors of insects by inducing changes in the synthesis of host volatiles? (3) How do symbiotic microorganisms act on insects by modulating behaviors?

## 1. Introduction

Insects are one of the most prosperous animals in the world [1]. In the course of long-term evolution, their olfactory system has been well-developed [2]. Currently, extensive studies demonstrate that olfaction serves as the primary sensory system of insects [3,4,5]. Odorant molecules in the natural environment penetrate into the pores of the olfactory sensillum to form the odorant molecule-odorant binding protein complex [6,7]. Then, they reach the olfactory receptors [8] and elicit changes in the action potentials. The axons of olfactory receptor neurons in the peripheral nervous system transmit signals to the antennal lobe, which acts as an olfactory “relay system” receiving inputs from olfactory sensory neurons [9], and then local interneurons connect olfactory neurons to projection neurons, which transmit signals to higher brain regions such as the mushroom body and lateral protocerebrum sites [10,11,12,13,14]. Finally, the brain issues instructions to regulate many critical behaviors, including locomotion, grooming, feeding, inter-species communication, reproduction, and host or prey selection. Collectively, the insect olfactory system is a sophisticated network of neurons that receive numerous forms of olfactory inputs via the peripheral and central neural systems, which are highly developed and deliberate [15,16]. Several types of neurons in the peripheral nervous system receives initial odor signals, and the signs are integrated through the central nervous system to modify insect behaviors [17]. Insects’ olfactory systems can sense and trace the source of the food odor to position themselves correctly for feeding, sense the odor of the living environment for gathering, recognize the sex pheromone to complete mating behaviors, and locate the spawning site to successfully lay eggs [18,19,20].

Microorganisms are widely present in insects’ habitats. Some microorganisms do not have active dispersal mechanisms of their own and need to use external forces, such as insect vectors, to help them spread [21,22,23]. In addition, a variety of microorganisms are nourished by insects, which coexist alongside their hosts [24], and many microorganisms also have remarkable influences on insect biological processes such as development [25], physiology [26], nutrition [27], survival [28], immunity [29], and even vector competence [30]. Thus, throughout the long period of co-evolution, insects have formed close relationships with microorganisms, which interact with and depend on each other, and evolve in concert [31,32,33,34]. Ample evidence suggests that certain volatiles released by microorganisms could have prominent influences on insect behaviors [35,36,37,38]. Additionally, chemical volatiles produced by microorganisms can also affect the host, which is closely associated with the reproduction and feeding behaviors of insects by regulating the olfactory preference of insects [39,40,41]. In addition, microorganisms can directly affect insects’ social behaviors through symbiotic relationships and participation in information exchange [42].

This article reviewed current understandings underlying the insect-microbial system, an exquisite network of interactions operating between the insect olfactory system and microorganisms, including fungi [43], bacteria [44], and viruses [45], and its future perspectives.

## 2. Volatiles Released by Microorganisms Directly Regulate Insect Behaviors through Olfactory Cues

Microorganism-derived volatile organic compounds (VOCs), which operate as insect semiochemicals and could have significant impacts on insect behaviors, are a variety of combinations of gas-phase, carbon-based molecules produced by microorganisms [46]. This section focuses on fungal and bacterial volatiles that directly alter insect behaviors through olfaction.

### 2.1. Effects of Fungal Volatiles on Insect Behaviors

Fungi can produce a variety of volatile chemicals, such as alcohols, ketones, acids, and esters, which could elicit attractive or repellent effects on insects, as well as physiological and behavioral responses of insects [47]. A previous study found that *Penicillium* was capable of producing typical volatiles, including geosmin, linalool, 3-carene and D-limonene, and that the same fungus elicited completely different reactions in the same species of insect. Non-gravid female *Bactrocera dorsalis* (oriental fruit fly, Hendel) and males exhibited robust aversive behavior toward *Penicillium*-inoculated substrates due to the presence of geosmin, while linalool, 3-carene and D-limonene could serve as efficient oviposition attractants for gravid female *B. dorsalis*, which could override the aversive activity of geosmin [25]. Thus, the odors from *Penicillium* differentially affected the behavioral responses and oviposition preference of *B. dorsalis* [25]. Additionally, a study also suggested that the microorganism-produced geosmin also triggers aversion of *Drosophila melanogaster*, thus geosmin is used by flies as a universal warning signal [48]. Geosmin activates only one class of sensory neurons that express the olfactory receptor Or56a. This neuron targets the DA2 glomerulus and transmits olfactory signals through axons to projection neurons. In *D. melanogaster*, DA2 activation by geosmin could override the inputs of other olfactory pathway signals that affect oviposition and feeding. Odorants produced by fungi themselves could impact the behavioral reactions of insects by affecting the input, transmission, and transfer of feedback of olfactory information [49]. Formosan subterranean termites, *Coptotermes formosanus*, exhibit aversive reactions to 3-octanone and 1-octen-3-ol, which are the major chemical compounds of *Isaria fumosorosea* K3 [50]. During the process of hunting for a food supply or an appropriate location for oviposition, mycophagous insects heavily rely on the chemical fingerprints released by fungal substrates, as the fruiting bodies and mycelia often produce certain volatiles. Female *Lycoriella ingenua* (fungus gnat) preferred uncolonized compost compared to colonized compost, as the presence of three active volatiles from colonized compost, including 1-hepten-3-ol, 3-octanone, and 1-octen-3-ol, could elicit antennal responses to result in clear avoidance towards these colonized sites [51]. These results from previous research imply that fungal volatiles could function to attract or repel insects, with the specific regulatory mechanisms varying according to the interactions between the fungi and the insect.

### 2.2. Effects of Bacterial Volatiles on Insect Behaviors through Olfaction

There is increasing evidence implying that typical volatile compounds from bacteria could also elicit changes in insect behaviors via olfactory cues [52]. There is evidence suggesting that bumblebees preferred to feed on nectar solutions inoculated with the bacterium *Asaia astilbes* over the yeast *Metschnikowia reukaufii* due to their preference for certain volatiles [53]. The authors suspected that such preference for the taste of *A. astilbes*-conditioned nectar may be driven by metabolites dissolved in nectar, such as acetic acid [53]. In the olive fly, *Bactrocera oleae*, electrophysiological studies demonstrated that antennal sensilla responded to *Pseudomonas putida* bacterial filtrate odor, methyl thioacetate, olive leaf and olive, while acetic acid produced an inhibitory response in single sensillum recording [54]. Similarly, a study has shown that bacterial metabolites also influence the egg-laying behavior of insects via their influences on olfactory recognition [55]. A study in the primary screwworm, *Cochliomyia hominivorax*, has shown that it prefers to oviposit on blood that has been inoculated by certain bacteria due to the volatile compounds released by these bacteria [56]. Further studies have found that five major volatiles from a mixture of five bacteria (*Klebsiella oxytoca* Flugge, *Proteus mirabilis* Hauser, *Proteus vulgaris* Hauser, *Providencia rettgeri* Rettger and *Providencia stuartii* Ewing), are attractive to *C. hominivorax* [56]. Gut bacteria also produce volatile substances that could trigger attractive or repellent behaviors in insects. Researchers found an interesting example: the honeydew produced by aphids contains volatile substances released by gut microorganisms, which could attract natural enemies and ants who live in symbiosis with the aphids [57,58]. These studies found that linalool and phenylethyl alcohol produced by *Staphylococcus xylosus* in the honeydew of *Aphis fabae* are important substances for the discrimination of symbiotic aphids by *Lasius niger*, as they could strongly attract *L. niger* in symbiosis to help protect the aphids and carbohydrate-rich honeydew for the ants [57,58]. These pieces of evidence suggest that microorganisms elicit attractive and repulsive effects on insects through their metabolites, and different species of insects use different behavioral strategies to respond to the volatiles of microorganisms (Table 1).

## 3. The Microorganism-Induced Changes in Host Volatiles Contributed to Altered Behaviors

Microorganisms’ action on plant hosts could alter the synthesis and emission of volatiles of host plants, resulting in certain degrees of changes in the olfactory preference of insects, and thus playing an important role in shaping interactions between plants and insect herbivores (Table 2).

### 3.1. Effects of Fungi-Induced Changes in Host Volatiles on Insect Behaviors through Olfaction

It is well-known that plants emit volatile blends, and the association of fungi with plants could alter the metabolic and physiological functions of the hosts, leading to a reshaped acceptability of the insects towards these plants [85,86]. *Arbuscular mycorrhizas* (AMs) change plant volatile emissions; specifically, AMs suppressed the emission of the sesquiterpenes (E)-caryophyllene and (E)-β-farnesene, and the attractiveness of aphids to plant VOCs was negatively associated with the proportion of these sesquiterpenes [65]. Consequently, AM-colonized broad bean (*Vicia faba*) was more attractive to aphids [65]. Many plants interact with AMs to increase nutrient acquisition, and herbivores such as aphids drain nutrients from plants. Both AM and aphids can affect plant metabolic pathways and may influence each other by altering the condition of the shared host plant [87,88]. It has been reported that this may be related to the aphid-induced defense signals involving the jasmonic acid and salicylic acid pathways [89,90]. Thus, insects and fungi formed tight associations by coordinating olfactory sensing and growth to minimize fitness costs. In addition, microorganism-induced changes in the volatiles of host plants are also essential for seeking egg-laying sites in insects. For instance, the polyphagous moth (*Helicoverpa armigera*) oviposited more on the leaves inoculated with *Acremonium strictum* compared to endophyte-free plants, especially between 10 days and 18 days following inoculation [66]. It is interesting to note that endophyte-inoculated plants emitted diverse terpenes and sesquiterpenes at significantly lower amounts as compared to endophyte-free plants. Additionally trans-β-caryophyllene was emitted significantly higher amounts in inoculated plants. Thus, the oviposition preference of moths may be an evolutionary adaptation to host plants, as the emissions of endophyte-inoculated plant volatiles help moths to escape from predators or parasitoids [66]. The spruce bark beetle, *Ips typographus*, is the most destructive forest pest in Europe. Behavioral experiments showed that immature adults of bark beetles are attracted to food sources colonized by their fungal symbionts, and this attraction is mediated by volatile cues released by colonized food [67]. Fungi colonize different parts of the plant, such as the rhizosphere and phyllosphere, to change the plant VOCs and thus could modify the olfactory responses of insects toward these plants.

### 3.2. Effects of Bacteria-Induced Host Volatiles on Insect Behaviors through Olfaction

Bacterial pathogens have also been shown to affect their host’s volatile and nonvolatile metabolites, which would lead to increased attraction toward insect vectors, thereby increasing the spread of these pathogens. Compared with aphid-infested plants without rhizobacteria, the volatile blends from rhizobacteria-treated aphid-infested plants are less attractive to the aphid parasitoid *Diaeretiella rapae*. The explanation for this phenomenon is that the rhizobacterium *Pseudomonas fluorescens* could modify the blends of aphid-infested plant volatiles by blocking the jasmonic acid pathway; thus, aphid-infested plants become less attractive for parasitoids [72]. This result supports the emerging idea that beneficial organisms do not benefit the plant in the defense against all attackers [72]. In addition to affecting locating, bacteria-induced changes in the volatiles of hosts also affect insect egg-laying. A study in *B. dorsalis* showed that an increased concentration of β-caryophyllene in the host fruits was related to the host-marking pheromone used by *B. dorsalis* to regulate the oviposition process, and such an increase was induced by the bacteria adhered to the laid eggs [73]. Apart from the aforementioned bacteria, there are also bacteria that colonize plant roots, namely, plant growth-promoting rhizobacteria (PGPR). *Bacillus amyloliquefaciens*, as a type of PGRP, causes plant volatiles to be altered when it is colonized on plant roots. Earwigs *Doru luteipes* (Dermaptera: Forficulidae), were attracted by plants that presented as PGPR-inoculated [91]. Bacteria form interactions with plants, which could alter physiological states, activate several metabolic pathways of plants, change the synthesis and emission of volatiles, and, consequently, reshape insect behaviors.

In addition to bringing influences on plant volatiles, bacterial pathogens also alter the synthesis of pheromones in insects, thus affecting their aggregation behaviors. Infection by pathogenic bacteria alters the social communication system of *Drosophila*. More specifically, infected *Drosophila* and their frass release significantly more fly odors, such as the aggregation pheromones methyl laurate, methyl myristate, and methyl palmitate, which attract healthy flies that later get infection and help spread the pathogen [76]. Such modulations in pheromone synthesis serve as another way to regulate insect behaviors to increase the fitness of microorganisms. In the German cockroach, *Blattella germanica*, the bacterial community in the gut is also essential for the synthesis of aggregation pheromones. Normal and gut bacteria-inoculated cockroach feces release extremely attractive volatile carboxylic acids that cause aggregation in cockroaches. In contrast, bacteria-free feces emit little volatile carboxylic acids and are less attractive for cockroach aggregation [77]. The gut microbial community contributes to the production of certain semiochemicals, which plays an important role in attracting insects to gather. However, a hygienic behavior also exists in insects and has developed as an important feature in controlling the increased risk of disease transmission that arises from sociality and group living [92,93]. European foulbrood is a globally distributed bacterial brood disease caused by *Melissococcus plutonius*, which could result in stunted development and even death of honeybee larvae. *M. plutonius* alters the chemical cues (brood pheromones, signature mixtures, diagnostic substances) emitted by honeybee larvae that play a central role in the sealing initiation of disease-specific behavior [78].

### 3.3. Effects of Virus-Induced Host Volatiles on Insect Behaviors through Olfaction

Viruses are widespread in both natural and agricultural plant communities, and plant viruses can manipulate their hosts to release odors that are attractive or repellent to their insect vectors [45]. The volatile compound chlorophyll alcohol released by *Southern rice black-streaked dwarf virus*-infected rice plants has a significant repellent effect on its vector, the white-backed planthopper (*Sogatella furcifera*), to repel them away from the infected plant [80,94]. The same genus of Fiji disease virus, *rice dwarf disease virus*, significantly promoted the release of (E)-β-caryophyllene and 2-heptanol from rice plants. Additionally, (E)-β-caryophyllene lured non-viruliferous green rice leafhoppers to settle on virus-infected plants [45]. A study showed that *cucumber mosaic virus* enhanced the attractiveness of aphids to these infected plants but reduced the quality of host plants to aphids due to the overall elevated volatile emissions from host plants caused by the virus [81]. Transmission of plant viruses by insect vectors is a complex and dynamic biological process involving a three-way interaction among plant viruses, insect vectors and host plants. Plant virus regulating volatiles of host plants could influence the olfactory responses of their insect vectors and thus largely facilitate virus spread.

In addition to plant viruses, arboviruses could also bring influences on insect behaviors. Zhang et al. showed that *dengue virus* and *Zika virus* infections led to a significant increase in the abundance of the skin microbiota in the infected individuals. These infected ones can metabolize and produce large amounts of acetophenone to activate the olfactory nervous system of mosquitoes, thereby significantly increasing their behavioral tendencies toward infected hosts. This phenomenon also favors rapid vector dispersal from infected insects [82]. A previous study also showed that infection with a virus causes changes in the olfactory preference of mosquitoes, as well as possibly modifying the oviposition site choice of female mosquitoes [95]. In conclusion, viruses also bring prominent effects on insect behaviors by regulating the metabolites of hosts (Table 2).

## 4. Symbiotic Microorganisms Regulate Olfaction in Insects

Symbiosis provides an opportunity for microorganisms to live together through mutual or one-way benefit with insects. Symbionts not only impact their hosts’ fitness but also fine-tune the olfactory system of insects [96,97,98]. Symbionts affect the olfactory learning and olfactory memory abilities of insects, influencing olfactory sensitivity and thus regulating insect behaviors. We also present cases of symbiotic microorganisms affecting insect olfaction.

### 4.1. Effects of Symbiotic Bacteria on Insect Behaviors through Olfaction

As one of the most widely studied microorganisms in symbiosis with insects, bacteria play irreplaceable roles in regulating the behaviors of insects. *Wolbachia* is the most common symbiotic bacterium in insects, which infects numerous insects and usually has negative influences on the reproductive system of insects. Peng’s [99] research revealed that *Wolbachia*-infected *Drosophila* might increase sensitivity toward food odors by having much higher transcript levels of the odorant receptor gene, the olfactory receptor co-receptor (*Orco*). Cai [100] found that enteropathogen infection in *Drosophila* can modulate fly olfaction through metabolic reprogramming of ensheathing glia of the antennal lobe. Such changes in the olfactory system contributed to promoting the avoidance of flies toward bacteria-laced food as well as fly survival [101]. The mushroom body of *Drosophila* is a neural circuit associated with memory formation and storage, which plays a key role in various behaviors [102]. Many researchers have investigated the effects of *Erwinia carotovora carotovora* (Ecc) on the olfactory nervous system of *Drosophila*. Babin [10] speculated that ingesting Ecc strengthened synaptic connections between the neurons that transport unconditioned stimulus information and the mushroom body Kenyon cells, which enhance olfactory-conditioned stimulus information in insects. Determined by Pavlovian conditioning assay and T-maze experiment, the Ecc-infected flies enhanced the selection ability toward 4-methylcyclohexanol, and 3-octanol compared to the control group (saline treated). However, Johanna [103] speculated that ingesting Ecc triggered dangerous signals for food sources to be directly or indirectly transmitted to the mushroom body, where an appropriate olfactory response could then be produced. Although flies sometimes prefer pathogenic bacterial volatiles and feed on them, their feeding is inhibited when infected. All these examples are sufficient to demonstrate that bacteria are capable of inducing changes in the olfactory nervous system of insects.

In addition, it has been found that gut bacteria themselves could bring direct effects on the olfactory system of insects. Zhang [104] demonstrated that gut *Lactobacillus* strains of honeybee could convert tryptophan into indole derivatives that stimulate the host aryl hydrocarbon receptor, thus promoting memory behaviors. In *Drosophila*, compared with conventional flies (i.e., with an unaltered microbiome), microbiologically sterile (axenic) flies displayed a moderate reduction in memory performance in olfactory appetitive conditioning [105]. This may be related to the activation of octopaminergic neurons in *Drosophila*, which reinforces olfactory memory in the mushroom bodies. Most insects bear microorganisms that influence their health and fitness, and these microorganisms tend to be harbored predominantly in the lumen of the gut [106]. From the current study, it appears that the symbiotic relationship between insects and gut microbiota affects the neural function and behaviors of the hosts as well [107]. The gut microbiota of *Drosophila* larvae was altered by treatment with antibiotics or probiotics, and the tropism response of the treated larvae to odorants was found to be reduced. It was concluded that the microbiota influences homeostatic mechanisms in the host that control Gamma-aminobutyric acid (GABA) production and GABA receptor expression, which are known to affect the olfaction of the host [107,108]. From the available results, it appears that gut bacteria could largely influence insect olfaction by affecting neural circuit communication.

### 4.2. Effects of Symbiotic Viruses on Insect Behaviors through Olfaction

Many viruses obtain access to the central nervous system (CNS) through peripheral sensory neurons via anterograde transfer [109]. The olfactory response of the primary carrier of a variety of flaviviruses, *Aedes aegypti* (Yellow fever mosquito), is susceptible to viral interference [110,111,112,113]. During *dengue virus* (DENV) infection, the expressions of genes related to the olfactory learning processes changed and thereby caused changes in the olfactory response [114,115,116]. For instance, DENV2 infection increases mRNA levels of *Ca^2+^/CaM serine/threonine kinase II*, a central molecule in mechanisms of synaptic plasticity and memory in the head of mosquitoes, resulting in a loss of olfactory preferences toward skatole in female mosquitoes [117]. *West Nile virus* induced changes in the host-seeking behaviors of mosquitoes by interfering with the mosquitoes’ central nervous system [118]. Thus, the virus itself could induce changes in the mosquitoes’ olfactory learning abilities and host-seeking behaviors.

In addition to the central olfactory nervous system, the peripheral nervous system may also be affected by the virus. In the case of plant viruses, rice stripe virus significantly increased *Orco* transcripts in *Laodelphax striatellus* (small brown planthopper, SBPH), resulting in stronger olfactory sensitivities and seeking abilities toward the odor of rice plant compared with healthy SBPHs [119]. *Spodoptera exigua multiple nucleopolyhedrovirus* (SeMNPV) is an entomopathogenic double-stranded DNA virus that primarily infects *Spodoptera exigua* (Beet armyworm) [120]. When larvae were infected with SeMNPV, the mRNA level of *Or35* was significantly upregulated and could recognize a wide range of odor molecules with different chemical structures faster compared to uninfected ones, especially toward aromatic hydrocarbons and terpenes [121]. The *deformed wing virus* (DWV) has been found to be able to affect the honeybee population. In addition to causing wing malformation and behavioral disturbances, DWV can be found in the antennal lobes of the brain, damaging the anatomical integrity of the infected area and impairing normal antennal function related to aroma perception. High viral loads reduced the gene expression of the odorant-binding protein on the antennas of middle- and forager-age bees, thus decreasing their sensitivities toward the odor of *Mentha piperita* [122]. Insects evolved a highly sophisticated olfactory system that allows them to perform complex behaviors. Virus infection could bring changes in the expressions of olfactory genes in insects, thus affecting olfactory sensitivity (Table 3).

## 5. Discussion and Future Perspectives

In this review, we summarized insect behavioral changes modulated by microorganism-induced alterations on olfactory cues (Figure 1). Insect’s olfactory system perceives volatiles, and microorganisms can induce changes occurring at any point in the process from receiving the signal to transmitting it to the brain to produce a response. In the insect-microorganism interaction process, insects absorb the nutrients they need for growth and development, and microorganisms promote their propagation through the insects, which facilitates their tight bindings. Certain compounds derived from microorganisms are potent repellent to insects, and they can help provide some protection against natural enemies, pathogenic infections, etc. And some compounds from microorganisms exhibit robust attraction toward insects. In addition, a myriad of studies found that microorganisms can not only influence insect behaviors by themselves but can also modify insect behaviors by interacting with insects or inducing dramatic changes in host volatiles. In the natural world, microorganisms can also exist on the body surface and in the internal body of plants and animals, which provides the opportunity to enable them to alter the physiological state and metabolic activities of their hosts, and consequently, to alter the synthesis and emission of host volatiles. Insects thus accordingly generate various behavioral responses by recognizing the characteristics of these volatiles and distinguishing changes in volatiles, including positioning, feeding, and egg laying. Therefore, based on existing studies, when insects form interactions with microorganisms, microorganisms are capable of acting directly on insect olfaction via their volatiles or indirectly modulation by inducing changes in host volatiles, as well as through their symbiotic interactions with insects. Apart from the effects of volatiles, microorganisms could also directly influence insect olfaction by altering the transmission process of the insect olfactory system and the expression of insect olfactory genes, thus affecting olfactory learning, memory, and sensitivity. Insect olfaction is a process in which the peripheral olfactory nervous system converts chemical signals into electrical signals, which are transmitted to the primary olfactory centers and then integrated and processed to the higher nerve centers, and finally triggering insect to initiate corresponding responses. The transmission of signals is a complex process in the central nervous system, and any changes in any steps will have corresponding effects, and microorganisms act as a link in this process, changing the integration, processing and transmission of olfactory signals.

With the gradual advancements of various studies, the research on the influences of microorganisms on insect olfaction has been expanded, and the improvement of various techniques has promoted the detection of volatile organic compounds produced by microorganisms that are more accurate and convenient, as well as the clear understanding of the regulatory mechanisms underlying insect olfactory system. But as far as we are concerned, most studies have focused on the effects of microorganisms or microorganism-induced changes in plant volatiles on insect behaviors. The related central olfactory processing and the corresponding molecular mechanisms are less studied. Moreover, the mechanisms by which microorganisms influence brain function are still incipient, and little is known about the related effects on the central nervous systems. In future work, a functional analysis of how external volatiles and olfactory nerves jointly affect insect olfaction is needed, and more attention should be paid to the neural olfactory regulation mechanism underlying microorganism-insect interactions. Using genomics and metabolomics to understand the interaction of microorganisms with insect olfaction, further biochemical and physical understanding of insect olfaction could be realized. Moreover, by using techniques such as molecular biology, neurophysiology, immunohistochemistry, and intracellular electrophysiology, in conjunction with tangible behavioral assays, the microorganisms modulated insect behaviors via olfaction could be analyzed from anatomical and physiological aspects, thus facilitating a better understanding of microorganism-host-insect interactions.

## Figures and Tables

**Figure 1 insects-13-01094-f001:**
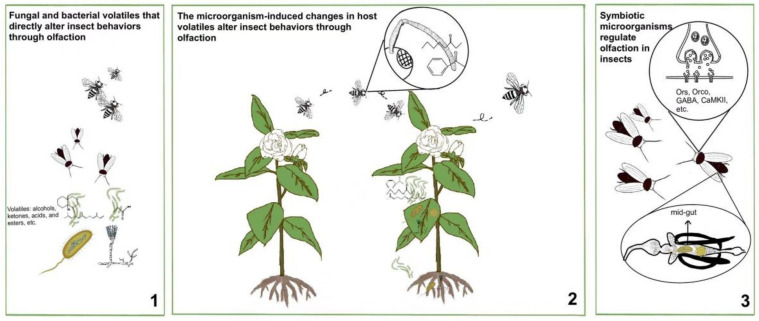
Microorganisms modulate insect behaviors via olfactory cues by different ways. 1 Fungal and bacterial volatiles that directly alter insect behaviors through olfaction. 2 The microorganism-induced changes in host volatiles alter insect behaviors through olfaction. 3 Symbiotic microorganisms regulate olfaction in insects.

**Table 1 insects-13-01094-t001:** Cases of microorganism volatiles trigger changes in insect behaviors through olfactory cues.

Microorganisms (Type)	Targets	Microbial Volatiles	Olfactory Modification	References
*Penicillium* (Fungi)	*Bactrocera dorsalis*	geosmin	Attracting egg laying	[25]
*Isaria fumosorosea* (Fungi)	*Coptotermes formosanus*	3-octanone and 1-octen-3-ol	Affecting direction of motion	[50]
*Agaricus bisporus* (Fungi)	*Lycoriella ingenua*	1-hepten-3-ol, 3-octanone and 1-octen-3-ol	Repelling egg laying	[51]
*Accharomyces cerevisiae* (Fungi)	*Drosophila melanogaster*	ethanol, acetic acid, acetoin, 2-phenyl ethanol and 3-methyl-1-butanol	Attracting egg laying	[59]
*Bacillus cereus* (Fungi)	*Ips paraconfusus*	2-Methyl-3-buten-2-ol	Facilitating aggregation behavior	[60]
*Fusarium solani* (Fungi)	*Drosophila melanogaster*	1-pentanol and 1-octen-3-ol	Attracting feeding	[61]
*uraishia capsulata, Scheffersomyces ergatensis, Peterozyma xylosa, Wickerhamomyces subpelliculosus, and Lachancea thermotolerans* (Bacteria)	*Bactrocera oleae* (Diptera: Tephritidae)	Isoamyl alcohol	Attracting mating	[62]
*Metschnikowia reukaufii* (Bacteria)	*Asaia astilbes*	2-ethyl-1-hexanol	Attracting feeding	[53]
*Pseudomonas putida* (Bacteria)	*Bactrocera oleae*	methyl thiolacetate, olive leaves and olives, α-pinene, acetic acid		[54]
*Klebsiella oxytoca Flugge, Proteus mirabilis Hauser, Proteus vulgaris Hauser, Providencia rettgeri Rettger and Providencia stuartii Ewing* (Bacteria)	*Cochliomyia hominivorax*	dimethyl disulfide, dimethyl trisulfide, phenol, p-cresol and indole	Attracting egg laying	[56]
*Staphylococcus xylosus* (Bacteria)	*Aphis fabae*	phenylethyl alcohol	Attracting feeding	[57,58,63]
*Corynebacterium sputi* (Bacteria)	*Aphidius colemani*	esters, organic acids, aromatics and cycloalkanes	Repelling parasitism	[64]

**Table 2 insects-13-01094-t002:** Cases of microorganism-induced host volatiles trigger changes in insect behaviors through olfactory cues.

Microorganisms (Type)	Targets	Microbial Volatiles	Olfactory Modification	References
*Arbuscular mycorrhizal* (Fungi)	*Acyrthosiphon pisum* L.	(E)-caryophyllene and (E)-β-farnesene	Attracting feeding	[65]
*Acremonium strictum* (Fungi)	*Helicoverpa armigera*	trans-β-caryophyllene	Attracting egg laying	[66]
*Endoconidiophora polonica, Grosmannia penicillata, Grosmannia europhioides, Ophiostoma bicolor* and Ophiostoma piceae (Fungi)	*Ips typographus*	3-methyl-1-butyl acetate, 2-methyl-1-butyl acetate	Attracting feeding	[67]
*Beauveria bassiana*, *Metarhizium acridum* (Fungi)	*Myzus persicae* and *Rhopalosiphum padi*	heptanal, octanal, nonanal and decanal	Attracting feeding	[68]
*Botrytis cinerea* (Fungi)	*Epiphyas postvittana*	Ethanol and 3-methyl-1-butanol	Repelling egg laying	[69]
*Diatraea saccharalis* (Fungi)	*Cotesia flavipes*	1-octen-3-ol	Repelling parasitism	[70]
*Beauveria bassiana* and *Metarhizium robertsii* (Fungi)	*Cosmopolites sordidus*	styrene, benzothiazole, camphor, borneol, 1,3-dimethoxy-benzene, 1-octen-3-ol and 3-cyclohepten-1-one	Repelling feeding	[71]
*Pseudomonas fluorescens* (Bacteria)	*Diaeretiella rapae*	jasmonic acid	Repelling parasitism	[72]
*Providencia*, *Klebsiella* (Bacteria)	*Bactrocera dorsalis*	β-caryophyllene	Attracting egg laying	[73]
*Erwinia carotovora carotovora* (Bacteria)	*Drosophila melanogaster*	methyl laurate, methyl myristate and methyl palmitate	Facilitating aggregation behavior	[74,75,76]
*Enterococcus avium, Weissella cibaria, Pseudomonas japonica, Pseudomonas monteilii, Acinetobacter pittii, Acinetobacter sp.* (Bacteria)	*Blattella germanica*	carboxylic acids	Facilitating aggregation behavior	[77]
*Melissococcus plutonius* (Bacteria)	*Apis mellifera*	cuticular hydrocarbon, brood ester pheromones, γ-octalactone	Repelling egg laying	[78]
*Candidatus Liberibacter asiaticus* (Las) (Bacteria)	*Diaphorina citri* Kuwayama	methyl salicylate	Attracting feeding	[79]
*Rice dwarf virus* (Virus)	*Nephotettix cincticeps*	(E)-β-caryophyllene and 2-heptanol	Attracting feeding	[45]
*Southern rice black-streaked dwarf virus* (Virus)	*Sogota furcifera*	Chlorophyll alcohol	Attracting feeding	[80]
*Cucumber mosaic virus* (Virus)	*Anasa tristis*	salicylic acid, ethylene	Attracting feeding	[81]
*Zika virus* (Virus)	*Aedes aegypti*	acetophenone	Attracting feeding	[82]
*Tomato yellow leaf curl virus* (Virus)	*Bemisia tabaci*	β-myrcene, thymene, β-phellandrene, caryophyllene, (+)-4-carene and α-humulene	Attracting feeding	[83]
*Cucurbit chlorotic yellows virus* (Virus)	*Bemisia tabaci*	terpenes	Attracting feeding	[84]

**Table 3 insects-13-01094-t003:** Cases of symbiotic microorganism trigger changes in insect behaviors through olfaction.

Microorganisms (Type)	Targets	Effects	Olfactory Modification	References
*Wolbachia* (Bacteria)	*Drosophila melanogaster*	Olfactory sensitivity	Attracting feeding	[99]
*Erwinia carotovora carotovora* (Bacteria)	*Drosophila melanogaster, Drosophila suzukii*	Olfactory sensitivity	Repelling feeding	[101]
*Pseudomonas entomophila* (Bacteria)	*rosophila melanogaster*	Olfactory sensitivity	Repelling feeding	[10,103]
*Lactobacillus plantarum* and *Acetobacter malorum* (Bacteria)	*Drosophila melanogaster*	Olfactory sensitivity	Repelling chemotaxis response	[123]
*West Nile virus* (Virus)	*Culex pipiens* Linnaeus	Olfactory memory	Repelling feeding	[118,124]
*Rice stripe virus* (Virus)	*Laodelphax striatellus*	Olfactory sensitivity	Repelling feeding	[119]
*Spodoptera exigua multiple nucleopolyhedrovirus*, *Autographa californica multiple nucleopolyhedro virus* (Virus)	*Spodoptera exigua*	Olfactory sensitivity	Attracting feeding	[121]
*Deformed Wings Virus* (Virus)	*Apis melllifera*	Olfactory sensitivity	Repelling feeding	[122]
*West Nile Virus* and *Bancroftian filarisis* (Virus)	*Culex quinquefasciatus*	Olfactory memory	Repelling feeding	[125]

## Data Availability

Not applicable.

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
