# Peer review of "Insect-Microorganism Interaction Has Implicates on Insect Olfactory Systems"

_insects, 2022, doi:10.3390/insects13121094_

Round 1

Reviewer 1 Report

Ai et al described the implication of microorganisms i.e bacteria, fungi, and viruses on insect olfaction. It seems interesting and will have of broad interest to insect readers. 

My suggestions are:

1. Please edit the manuscript for English editing and scientific language editing.

2. I would suggest drawing a figure or two, which could describe the whole idea of the MS. graphical illustration would be better for the readers.

Author Response

Ai et al described the implication of microorganisms i.e bacteria, fungi, and viruses on insect olfaction. It seems interesting and will have of broad interest to insect readers. 

My suggestions are:

  1. Please edit the manuscript for English editing and scientific language editing.

Response: We thank for this reviewer’s suggestion. The manuscript has been sent to MDPI Language Editing Service for language editing with the order number of english-edited-53962 and we have corrected the syntax and grammar errors throughout the full text. We hope it could meet the publication standard of the journal.

I would suggest drawing a figure or two, which could describe the whole idea of the MS. graphical illustration would be better for the readers.

Response: We thank this reviewer for his/her good suggestion. We drafted the graphic abstract-like image (Figure 1) to summarize the whole idea and provide in the revised manuscript.

Reviewer 2 Report

In this manuscript, Ai et al. discuss the interaction between insects and microorganisms – fungi, bacteria, and viruses- specifically focusing on how the volatile compounds released by the microorganisms modulate insect olfactory behaviors. This review details the influence of microorganisms-produced volatiles, microorganisms-induced changes in host volatiles, and symbiotic microorganisms on insect olfactory responses, including learning and memory.  

Recent studies in various insect species demonstrated a stronger influence of microorganisms on insect olfaction, which play a crucial role in locating food, mating partners, threats, oviposition sites, etc. Given the vast reports, this review is timely and of general interest to the readership of the Insect journal. However, the article is limited by clarity and a comprehensive topic description. Here are a few suggestions for the authors to consider

Major comments:

1. Throughout the manuscript, the authors state the volatile compound produced by microorganisms and its effect on insect olfaction. Discussing the ecological and evolutionary implications of these microorganism-insect interactions is important. For example, in lines 80-87, substrates containing geosmin produced by fungi act as an attractant for some insects, whereas repellent for others for oviposition preference. How do the same compounds induce contrasting behaviors, what are the advantages/disadvantages, etc, that need to be discussed?

2. This review article focuses specifically on the influence of volatiles on insect olfaction. However, the authors fail to provide specific details of olfactory behaviors (example: increased repulsion to odors), instead use a generic term for olfactory behaviors such as sensitivity, memory, and response. For instance, “….influenced the olfactory behavior” (line 211); “…changes in olfactory behaviors” (line 286), etc. 

3. Tables list the microorganisms, volatile compounds, and target insect details. It would be useful if the author included specific olfactory modifications in target species. In both tables, the second column can be merged with the first one, providing more information about olfactory behaviors.

4. Huge scope to improve the clarity and flow of the content. For example: 

a. “Collectively, the insect olfactory system is a sophisticated network of neurons that receive……..” (lines 43- 47). The insect olfactory system is very well characterized. For readability,  authors should state the specific neuronal subtypes that refer to (peripheral: olfactory receptor neurons, central: mushroom body, lateral horn, neuroendocrine, etc). 

b. Lines: 103-105, unclear sentence. 

c. Rather than the context, the authors have randomly used the words - olfactory plasticity, inhibitory response, etc.

5. As stated above, compared to other sensory circuits insect olfactory system is relatively well characterized in terms of circuit and function. The discussion material states, “how microorganisms act on the olfactory nerves system of insects needs to be investigated” which gives an impression that the circuit and molecular mechanisms remain completely unclear. To quote, geosmin dedicated olfactory circuit and odor-induced behavioral modification reported in Drosophila. Authors should discuss the olfactory circuits, repertoire, downstream mechanisms, and evolutionary significance of microorganism and insect olfactory interactions. 

Minor comments:

1. Line 42: Specify insect higher brain centers that receive olfactory signals.

2. Line 43; “….many critical behaviors”. What are the behaviors?

3. Line 20: “…messaging” Is it referred to interspecies communication?

4. Line 52: “In the habitat of the insect, microorganisms are ubiquitous.” It gives the impression that microorganisms are not ubiquitous in other habitats.

5. For easy readability, in addition to the scientific name, provide the common name of the organisms.

6. Line 243: “………the critical odorant receptor gene olfactory receptor co-receptor”. Authors refereeing to Orco here?

7. Throughout the manuscript, follow standards for the usage of expansions and abbreviations. 

Author Response

Reviewer #2 

 In this manuscript, Ai et al. discuss the interaction between insects and microorganisms – fungi, bacteria, and viruses- specifically focusing on how the volatile compounds released by the microorganisms modulate insect olfactory behaviors. This review details the influence of microorganisms-produced volatiles, microorganisms-induced changes in host volatiles, and symbiotic microorganisms on insect olfactory responses, including learning and memory.  

Recent studies in various insect species demonstrated a stronger influence of microorganisms on insect olfaction, which play a crucial role in locating food, mating partners, threats, oviposition sites, etc. Given the vast reports, this review is timely and of general interest to the readership of the Insect journal. However, the article is limited by clarity and a comprehensive topic description. Here are a few suggestions for the authors to consider

Major comments:

  1. Throughout the manuscript, the authors state the volatile compound produced by microorganisms and its effect on insect olfaction. Discussing the ecological and evolutionary implications of these microorganism-insect interactions is important. For example, in lines 80-87, substrates containing geosmin produced by fungi act as an attractant for some insects, whereas repellent for others for oviposition preference. How do the same compounds induce contrasting behaviors, what are the advantages/disadvantages, etc, that need to be discussed?

Response: We thank this reviewer for his/her suggestion. The study reported that geosmin signaled the presence of Penicillium and acted as repellent to non-gravid and male B. dorsalis, in the meanwhile, the blends from Penicillium, including linalool, 3-carene or D-limonene, could be sufficient to elicit attraction in gravid female to override such aversive from geosmin. We are sorry for such unclear information and cause misunderstandings, and we have rephrased this sentence in the manuscript.

  1. This review article focuses specifically on the influence of volatiles on insect olfaction. However, the authors fail to provide specific details of olfactory behaviors (example: increased repulsion to odors), instead use a generic term for olfactory behaviors such as sensitivity, memory, and response. For instance, “….influenced the olfactory behavior” (line 211); “…changes in olfactory behaviors” (line 286), etc. 

Response: We thank this reviewer for his/her suggestion and we apologized for our oversight not to provide clear information. We provided specific detail descriptions underlying behavioral changes in the revised manuscript. We are sorry for our negligence, and we have added more information underlying their behavioral changes.  

  1. Tables list the microorganisms, volatile compounds, and target insect details. It would be useful if the author included specific olfactory modifications in target species. In both tables, the second column can be merged with the first one, providing more information about olfactory behaviors.

Response: We thank this reviewer for his/her suggestion. We have merged two columns and added specific olfactory modifications in the revised version.

  1. Huge scope to improve the clarity and flow of the content. For example: 
  2. “Collectively, the insect olfactory system is a sophisticated network of neurons that receive……..” (lines 43- 47). The insect olfactory system is very well characterized. For readability,  authors should state the specific neuronal subtypes that refer to (peripheral: olfactory receptor neurons, central: mushroom body, lateral horn, neuroendocrine, etc). 

Response: We apologize for our oversight and making such mistake. We have added detailed descriptions underlying those specific neuronal subtypes and their functions in olfactory recognition processes as this reviewer kindly suggested.

  1. Lines: 103-105, unclear sentence. 

Response: We apologize for our oversight. We have rephrased this sentence in the revised manuscript.

  1. Rather than the context, the authors have randomly used the words - olfactory plasticity, inhibitory response, etc.

Response: We apologize for our oversight and making such mistake. We will be more careful when we use the term of “olfactory plasticity. And for the term of “inhibitory response: was originated from the reference we cited. To keep consistency, we used the term “inhibitory response” as in the original report. As this reviewer kindly suggested, this sentence might cause misunderstanding. Thus we added specific olfactory modifications following “inhibitory responses” in the revised manuscript. We thank this reviewer for his/her good suggestion.

Liscia, A.; Angioni, P.; Sacchetti, P.; Poddighe, S.; Granchietti, A.; Setzu, M.D.; Belcari, A. Characterization of olfactory sensilla of the olive fly: Behavioral and electrophysiological responses to volatile organic compounds from the host plant  and bacterial filtrate. J. Insect Physiol. 2013, 59, 705-716, doi: 10.1016/j.jinsphys.2013.04.008.

  1. As stated above, compared to other sensory circuits insect olfactory system is relatively well characterized in terms of circuit and function. The discussion material states, “how microorganisms act on the olfactory nerves system of insects needs to be investigated” which gives an impression that the circuit and molecular mechanisms remain completely unclear. To quote, geosmin dedicated olfactory circuit and odor-induced behavioral modification reported in Drosophila. Authors should discuss the olfactory circuits, repertoire, downstream mechanisms, and evolutionary significance of microorganism and insect olfactory interactions. 

Response: Thank you for your comments and apologize for our oversight. We deleted this sentence to avoid misunderstanding as this reviewer kindly suggested. And we also added prospect underlying the effects of microorganisms on olfactory circuits in the revised manuscript and geosmin-mediated olfactory circuit processing.

Minor comments:

  1. Line 42: Specify insect higher brain centers that receive olfactory signals.

Response: Thank you for your comments. We added detailed descriptions underlying insect higher brain centers that receive olfactory signals in the revised manuscript.

  1. Line 43; “….many critical behaviors”. What are the behaviors?

Response: Thank you for your comments. We already specify these related behaviors in the revised manuscript. The behaviors include habitat selection, access to food, avoidance of predators, inter-species communication, aggregation, and reproduction.

  1. Line 20: “…messaging” Is it referred to interspecies communication?

Response: Thank you for kindly suggestion. To avoid misunderstanding, we replaced “messaging” with the term of “interspecies communication”.

  1. Line 52: “In the habitat of the insect, microorganisms are ubiquitous.” It gives the impression that microorganisms are not ubiquitous in other habitats.

Response: We thank this reviewer for his/her suggestion and we apologized for our oversight not to provide clear information. To avoid misunderstanding, we have rephrased this sentence in the manuscript.

  1. For easy readability, in addition to the scientific name, provide the common name of the organisms.

Response: We are deeply sorry for our negligence. We have added the common name of the organisms in the revised manuscript.

  1. Line 243: “………the critical odorant receptor gene olfactory receptor co-receptor”. Authors refereeing to Orco here?

Response: We apologize for our oversight. Yes, we are referring the Orco and we added “Orco” in the revised manuscript to make it clear.

  1. Throughout the manuscript, follow standards for the usage of expansions and abbreviations. 

Response: We apologize for our negligence. We have checked thorough for the abbreviations and correct for the misusage in the revised manuscript.

Reviewer 3 Report

Authors address a very interesting topic that of VOC mediated microbes-insects interactions. It is a vast chapter in Biological sciences and difficult to be addressed sufficiently in just a few pages.

The current article is well written and summarizes adequately (to an extent) the issue at hand providing enough starting points to concerned reader for further research.

I think the manuscript can be accepted if the authors give more clear and direct answers to the following comments:

L19. "olfactory" -> "olfactory cues" or "olfaction"

L35. "hearing and vision tend to degenerate" : [2]does not imply this.

L74. "are contributed" -> "contribute"

L107-109. [54] "inhibitory response" refers to SSR recording and not behavioral assays. Please clarify as it might be misleading to the readers.

L112. "spawn": [Spawn is the eggs and sperm released or deposited into water by aquatic animals.] - maybe it would be better to use a different word

L128. "olfactory behaviors": the terms bear no meaning as olfaction is not a behavior. Consider rephrasing as "behaviors modulated by olfactory cues"

L130. similar to above. Alternatively provide a clear explanation of the term.

L158. "by using": do authors mean maybe "who use"? (the parasitoids use volatiles as cues).

L170. "affected": "affecting"?

L169-171. correct but somehow unclear, please rephrase and specify.

L176. "to host-marking pheromone of B. dorsalis": As beta-caryophyllene is not produced either by larvae or adults B. dorsalis the correct term would be "host-marking semiochemical used by B. dorsalis".

L190. “illness”: consider “pathogen”

L195.  “are generally unpleasant”: unpleasant to whom? Please rephrase

L201.  Please add the explanation of Melissococcus plutonius (agent of European foul brood, an important disease of honey bees) for the less familiar readers.

L211. reconsider the term “olfactory behavior”

L242. “increase olfactory responses”: = “behavioral responses mediated by olfactory cues”

L314. “regulation ways”? Please specify and rephrase

Author Response

Reviewer #3

Authors address a very interesting topic that of VOC mediated microbes-insects interactions. It is a vast chapter in Biological sciences and difficult to be addressed sufficiently in just a few pages.

The current article is well written and summarizes adequately (to an extent) the issue at hand providing enough starting points to concerned reader for further research.

I think the manuscript can be accepted if the authors give more clear and direct answers to the following comments:

L19. "olfactory" -> "olfactory cues" or "olfaction"

Response: Thank you for your comments, we apologized for our oversight. As the reviewer indicated, we have corrected the word.

L35. "hearing and vision tend to degenerate" : [2]does not imply this.

Response: We are sorry for quoting the wrong reference. As kindly pointed by this reviewer, such statement might not be accurate. We deleted this sentence to avoid misunderstanding.

L74. "are contributed" -> "contribute"

Response: Thank you for your comments, we apologized for our oversight. As the reviewer indicated, we have corrected the word.

L107-109. [54] "inhibitory response" refers to SSR recording and not behavioral assays. Please clarify as it might be misleading to the readers.

Response: We are sorry for our negligence. As pointed by this reviewer, we have rephrased this sentence in the revised manuscript.

L112. "spawn": [Spawn is the eggs and sperm released or deposited into water by aquatic animals.] - maybe it would be better to use a different word

Response: Thank you for your comments. As kindly pointed by this reviewer, such statement might not be accurate. We have rephrased this sentence in the revised manuscript.

L128. "olfactory behaviors": the terms bear no meaning as olfaction is not a behavior. Consider rephrasing as "behaviors modulated by olfactory cues"

Response: As kindly pointed by this reviewer, such statement might not be accurate. We have rephrased this sentence in the revised manuscript.

L130. similar to above. Alternatively provide a clear explanation of the term.

Response: As kindly pointed by this reviewer, such statement might not be accurate. We have rephrased this sentence in the revised manuscript.

L158. "by using": do authors mean maybe "who use"? (the parasitoids use volatiles as cues).

Response: We are sorry for our unclear sentence. We have rephrased this sentence in the revised manuscript.

L170. "affected": "affecting"?

Response: Thank you for your comments, we apologized for our oversight. As the reviewer indicated, we have corrected the word.

L169-171. correct but somehow unclear, please rephrase and specify.

Response: Thank you for your comments, we apologized for our oversight. We have rephrased this sentence in the revised manuscript.

L176. "to host-marking pheromone of B. dorsalis": As beta-caryophyllene is not produced either by larvae or adults B. dorsalis the correct term would be "host-marking semiochemical used by B. dorsalis".

Response: Thank you for your comments, and we apologized for such unclear information. We have rephrased this sentence in the revised manuscript.

L190. “illness”: consider “pathogen”

Response: Thank you for your comments, we apologized for our oversight. As the reviewer indicated, we have corrected the word.

L195.  “are generally unpleasant”: unpleasant to whom? Please rephrase

Response: Thank you for your comments, we apologized for our oversight. We have rephrased this sentence in the revised manuscript.  

L201.  Please add the explanation of Melissococcus plutonius (agent of European foul brood, an important disease of honey bees) for the less familiar readers.

Response: Thank you for your comments, we apologized for our oversight. We added detailed explanation underlying Melissococcus plutonius in the revised manuscript.

L211. reconsider the term “olfactory behavior”

Response: Thank you for your comments, we apologized for our oversight. We revised this word in the manuscript to be clearer.

L242. “increase olfactory responses”: = “behavioral responses mediated by olfactory cues”

Response: Thank you for your comments, we apologized for our oversight. We have rephrased this sentence in the revised manuscript. 

L314. “regulation ways”? Please specify and rephrase

Response: Thank you for your comments, we apologized for our oversight. We have rephrased this sentence in the revised manuscript. 

Round 2

Reviewer 2 Report

The authors have satisfactorily addressed the reviewer's comments in the revised manuscript. Here are some suggestions for the authors to consider. While the authors made a significant improvement in the current version, there is still scope to improve the flow, and clarity of the review. For instance "........and improve learning memory of aversive odors." (Line 314). Authors should reconsider using the terms like "promoting memory behavior", "repelling positioning", and "attracting aggression", etc and the manuscript needs to be copy-edited for typographical errors (for example, ORco should be Orco). Finally, figure 1 can be better presented by summarizing the article. The current version of the figure is incomplete and pixilated, also limited with information both in the text and legend.  

Author Response

Dear editors and reviewers,

Thank you very much for your letter and for all the comments on our manuscript entitled “Insect-microorganism interaction has implicates on insect olfactory” (insects-1989413). All of the comments and suggestions are very helpful for improving our paper. We have studied the comments carefully and made corrections which we hope to meet with approval. Listed below are our responses to the editors and reviewers’ comments, presented in a point-by-point manner under the original reports. We also provided a manuscript with tracked changes to make sure the editors and reviewers can readily assess the changes we have made.

Reviewer’s comments

The authors have satisfactorily addressed the reviewer's comments in the revised manuscript. Here are some suggestions for the authors to consider. While the authors made a significant improvement in the current version, there is still scope to improve the flow, and clarity of the review. For instance "........and improve learning memory of aversive odors." (Line 314). Authors should reconsider using the terms like "promoting memory behavior", "repelling positioning", and "attracting aggression", etc and the manuscript needs to be copy-edited for typographical errors (for example, ORco should be Orco).

Response: We thank this reviewer for his/her suggestion and we apologized for our oversight. Drosophila infected with intestinal pathogens showed enhanced aversive olfactory learning compared to uninfected controls, which was assessed by a Pavlovian conditioning assay (Babin et al., 2014). And the authors concluded that flies intestinally infected with the mildly virulent Ecc showed 21% better learning performance than saline controls for 1 h retention. To avoid misunderstanding, we have rephrased this sentence in the manuscript.

Babin, A.; Kolly, S.; Kawecki, T.J. Virulent bacterial infection improves aversive learning performance in Drosophila melanogaster. Brain Behav. Immun. 2014, 41, 152-161, doi: 10.1016/j.bbi.2014.05.008.

Finally, figure 1 can be better presented by summarizing the article. The current version of the figure is incomplete and pixilated, also limited with information both in the text and legend.  

Response: We thank this reviewer for his/her suggestion. We redrafted figure 1 to better summarize the article.